

**Appraisal on inversion algorithm techniques in 2D electrical resistivity tomography**
**survey data for poised mapping of subsurface features**
Abhay Kumar Bharti*, Amar Prakash and Krishna Kant Kumar Singh
CSIR-Central Institute of Mining and Fuel Research, Dhanbad, India- 826001
*Corresponding author's email: abhay.bharati@gmail.com
**Abstract:**
Analysis of non-uniqueness model in resistivity imaging data is vital in inaugurating the
consistency of models. Nevertheless, such analysis is moderately unusual in resistivity
imaging data set. Electrical resistivity tomography (ERT) technique is being constantly used
in many scientific areas including engineering, environmental and archaeological survey.
Primarily, the inversion algorithm techniques are employed on synthetic model data set with
and without some random Gaussian noise, and its validity is tested by filed data set. The
study was conducted in the premises of Central Institute of Mining and Fuel Research
(CIMFR), Dhanbad by laying an ERT profile of 480 m length with 5 m electrode spacing
using Syscal Pro (Iris instrument) resistivity meter. Two standard arrays were used in this
study namely Wenner-Schlumberger and dipole-dipole array. The data set was mixed to a
single array to achieve better resolution and enhanced clarification. On processing data by
Prosys-II software, it was exported in Res2Dinv software for inversion. In this context, data
was inverted by different algorithm techniques i.e. least square ($L_2$-norm) and robust
inversion ($L_1$-norm). Exemplary results related to the heterogeneity of the resistivity structure
within the high and low resistivity anomaly were obtained by robust inversion method. The
obtained results are in broad agreement with the simulation model.
***Keywords:*** *ERT, Least square, Robust inversion, CSIR-CIMFR, Existing structures*




## 1.0 Introduction:

A significance of ground based geophysical technique is related to as much information as imaginable on subsurface existing structures and composition of materials which remain in the subsoil. All surface geophysical techniques are function of physical properties of the earth materials. Changes in subsurface properties such as porosity, permeability, density, saturation of water etc. may be distinguished by geophysical survey like gravity, seismic, and electrical methods (e.g., Ezersky 2008; Keller and Frischknecht 1996). Currently, it is common to apply geophysical techniques to environmental, engineering and mining related problems at shallow depths and it is valid solution for target identification at complex subsurface structures.

A non-invasive surface geophysical technique such as Electrical Resistivity Tomography (ERT) belongs to the family of the most applied geophysical methods in an extensive spectrum of mapping of near surface problems and environmental studies (e.g., Singh et al., 2004; Dahlin and Zhou 2006; Chandra et al., 2008; Kumar 2012; Singh 2013b; Bharti et al., 2016a,b; Bharti et al., 2019). This technique has become the most routinely used geoelectrical application for delineating the complex geological features of subsurface of the earth due to its comparative effortlessness and time effectivity. A better understanding of the subsurface geoelectrical structures in hard rocky terrain can be achieved by this technique. Electrical properties of the earth mass at shallow depth can be obtained by 2D ERT technique in both vertical and horizontal orientations, which helps in notching up of status of strata in qualitative and quantitative forms.

The geoelectric distinction between dissimilar natures of earth constituents is a reasonable means to classify different material characteristics, which has been allocated to the degree of weathering, moisture content and mineralogical composition of such earth material. A cell-based inversion technique is usually implemented for effectively prototypical complicated



structures along an uninformed resistivity spreading in subsurface of the earth. Therefore, this
procedure makes numerous rectangular cells with fixed positions and sizes by division of
subsurface block.
A few researchers have done work over comparison of inversion techniques in 2D resistivity
data sets. For example, the work done by Loke (2003), encompasses analysis on smooth and
blocky inversion methods in 2D resistivity survey. According to them, better results are
obtained by smooth inversion method in which change in resistivity is gradual; on the other
hand outcomes of blocky inversion method gives significant results for sharp boundaries. The
study was carried out over karstic structures by Hamdan and Vafidis (2009), by inversion
techniques for eminent image of resistivity. Three different inversion methods i.e. combined,
smoothness constrained and robust inversion were adopted on real data set and results were
compared and also combined inversion of two standard configurations namely, Wenner-
Schlumberger and dipole-dipole was conducted to obtain the highest reliability of the 2D
resistivity section.
The CSIR-CIMFR campus situated in Dhanbad area, India. Dhanbad, the coal capital of
country, lies in the mid-eastern part of Jharkhand state. Dhanbad district is evaluated as dry
because of deficiency of immense rivers and high temperature. The district is related to small
scale of ponds and two big dams which are good medium to recharge groundwater.
Therefore, groundwater arises in this zone below unconfined state in the weathered
formations at low depths in utmost of the lithological components in the Achaeans and nearly
all the lithological components in the Gondwana formation. Groundwater arises below
confined to semi confined state where the fractures are deep seated and are unrelated with
the top weathered formation (e.g., Kumar 2018).




## 2.0 Methodology:

In general, inversion procedure is involved to renovate the real circulation of acquired resistivity data sets as the latter does not deliver anticipated facts. The present study was conducted over the premises of Central Institute of Mining and Fuel Research, Dhanbad, India as shown in Fig. 1. Keeping in view the literature review in background, the scope of study encases analysis of two different inversion methods, namely, least square ($L_2$-norm) and robust inversion ($L_1$-norm) in 2D resistivity data set for mapping of complex subsurface existing structures. The idea of multiple inversion techniques could be used for evaluating the superiority of true 2D resistivity models. Inversion technique is a procedure to create a model that clarifies a set of measurements. It is related to make direct assumptions about the earth from DC resistivity measurements due to the contests of envisaging large data sets (e.g., Loke et al., 2003).

## 2.1 Synthetic Model:

Initially, the comparison of two different algorithm techniques i.e. least square and robust inversion in 2D electrical resistivity data set to map the complex subsurface existing structures through forward modelling, considered for better interpretation of field data set using RES2DMOD software package (e.g., Loke and Barker, 1996).

In this context, the model was consisted of four homogeneous layers i.e. (i) soil/alluvium layer, (ii) semi weathered rock layer, (iii) hard weathered rock layer and (iv) bed rock/ basement rock layer where their apparent resistivity values are of 100 Ωm, 300 Ωm, 500 Ωm, and 1000 Ωm with 64 equally spaced electrodes with 5m interval using finite difference algorithm technique. Finite difference algorithm technique divides the model subsurface into a number of rectangular blocks (e.g., Loke et al., 2003). Two conductive body (resistivity ranging 10 Ωm to 100 Ωm) and one resistive body (resistivity ranging 1000 Ωm to 2000 Ωm) was incorporated in model set.



The simulated resistivity retorts of the section were initiated using Wenner-Schlumberger,
dipole-dipole and combined inversion of both arrays with and without some random Gaussian
noise added to validate field condition and get more representative results. The synthetic
apparent resistivity model data set was inverted by using RES2DINV software for producing
true resistivity variation of subsurface of the earth.
Figure 2 and 3 shows the obtained outcomes from Wenner-Schlumberger, dipole-dipole and
combined inversion of both arrays using least square and robust inversion algorithm
techniques. Stimulatingly, all outcomes recover the anomaly locations through both inversion
techniques. However, in robust inversion technique was recognized both depth and
extensions of anomaly in all inverted resistivity models with greater resolution compared to
least square technique.  It is also observed that the combined inversion of both arrays gives
the better results with high resolution compared to Wenner-Schlumberger and dipole-dipole
array. For example, Dahlin and Zhou (2004), reported that the imaging with combined arrays
generates models similar to the preferable observation model among the specific array.
**2.2 Smooth-constrained least-squares technique:**
This technique usually uses the form of regularised least-squares optimization method in the
smooth-constrained or $L_2$-norm. The mathematical expression of this technique (e.g.,
deGroot-Hedlin and Constable 1990; Ellis and oldenburg 1994) is expressed as:
$$(J_i^{\ T} J_i + \lambda_i W^T W)\,\Delta r_i = J_i^{\ T} g_i - \lambda_i W^T W r_{i-1} \qquad (1)$$
Where, $g_i$ = data misfit vector,

$\Delta r_i$ = change in the model parameters for the ith iteration,

W= roughness filter,

$\lambda$ = damping factor,

$r_{i-1}$ = model parameters vector for the previous iteration and

J = Jacobian matrix of partial derivatives.



Roughness is filtered by first-order finite difference operator (e.g., deGroot-Hedlin and
Constable 1990). The equation (1) helps in minimising the sum-of-squares of the data misfit
and sum-of-squares of the model roughness.
Smooth-constrained least-squares technique reduces the sum of squares of the spatial changes
in the model resistivity and the data misfit. Optimal results are obtained for geologically
smooth variation subsurface (e.g., Barker 1992). However, it shows spread boundaries for
sharp transition like igneous dyke.
**2.3 Robust or blocky inversion technique:**
The cumulated absolute value of spatial changes in resistivity model can be reduced by
Robust inversion technique. It is also known as $L_1$-norm measure of the data misfit (e.g.,
Claerbout and muir 1973). The mathematical formulations used by $L_1$-norm optimisation
method is
$$(J_i^T R_d J_i + \lambda_i W^T R_m W) \Delta r_i = J_i^T R_d g_i - \lambda_i W^T R_m W r_{i-1} \qquad (2)$$
Where, $R_d$ and $R_m$ = weighting matrices
Constant resistivity values of each part are produced on application of $L_1$-norm to model
roughness filter (e.g., Farquharson and Oldenburg 1998). Sharp boundary separation is also
obtained by this technique.
**3.0 Discussions:**
2D ERT section of profile AA' was generated by the configurations of Wenner-
Schlumberger, dipole-dipole and combined inversion of both arrays for the length of 480 m
with electrode interval of 5m using Syscal Pro (Iris instrument) resistivity meter with 96
electrodes (Fig.2). Least square and robust inversion technique was adopted for analysis of
subsurface existing geological formation using Res2Dinv handling software as shown in Figs.
3 & 4.



**3.1 Inverted geoelectrical section of Least square inversion of profile AA'**
The 2D geoelectric model of profiles AA' along with the least square inversion technique
projected using Wenner-Schlumberger, dipole-dipole and combined inversion of both arrays
are shown in Figures 3a, b & c respectively.  The outcomes obtained by electrical resistivity
tomography designate an extensive range of resistivity variation through the profile.
Topmost layer up to a depth of 10m consisting of soil/ alluvium having a resistivity of about
2 to 80 Ωm was considered for all ERT sections. Two water aquifers ($L1Z2^{ws}$ & $L2Z2^{ws}$)
associated with fracture zones with relatively low resistivity of 2 to 12 Ωm at the surface
distance of about 130 m to 180 m and 280 m to 305m were delineated in 2D geoelectric
section generated by Wenner-Schlumberger array ( Fig.3a) and one water body ($L2Z2^{c}$)  was
also identified in combined inversion of both arrays along the surface distance at about 280 m
to 305m ( Fig.3c).  Relatively high resistivity (230 to 608 Ωm) anomaly associated with
weather rock / fracture rock ($WZ2^{dd}$ & $WZ2^{c}$) was identified along 2D ERT section of dipole-
dipole and combined inversion of both arrays at reduced distance (RD) of 25 to 90 m (Fig.3b
& c). A high resistivity contrast of more than 1600 Ωm associated with bed rock/ hard rock
($HZ3^{ws}$, $HZ3^{dd}$ & $HZ3^{c}$) was detected in all 2D resistivity sections (Wenner-Schlumberger,
dipole-dipole and combined inversion of both arrays) along a surface distance of 215 to 280
m.
**3.2 Inverted geoelectrical section of Robust inversion of profile AA'**
2D ERT inverse model of profiles AA' along with the robust inversion technique projected
using Wenner-Schlumberger, dipole-dipole and combined inversion of both arrays are shown
in Figures 4a, b & c respectively.  Wide range in resistivity was observed by this technique
also.
Top layer consisting of soil/ alluvium was encountered up to depth of 10 m followed by
Wenner-Schlumberger, dipole-dipole and combined inversion of both arrays are shown in


Figures 4a, b & c respectively. A prominent signature of relatively low resistive (L2Z2$^{ws}$,
L2Z2$^{dd}$, L2Z2$^{c}$) water aquifer zone associated with fracture rock mass at RD of about 280 m
to 305 m was identified with resistivity range of about 2 to 80 Ωm in all 2D ERT section
models. In addition, one water body (L2Z2$^{ws}$) was also identified in Wenner-Schlumberger
array along the surface distance of about 280 m to 305m ( Fig.4a). A signature of weather
rock / fracture rock (WZ1$^{ws}$, WZ1$^{dd}$ & WZ1$^{c}$) was recognized along 2D ERT sections of
Wenner-Schlumberger, dipole-dipole and combined inversion of both arrays at RD of 25 to
110 m with moderately high resistivity range of 230 to 608 Ωm (Fig.4). The bed rock/ hard
rock (HZ3$^{ws}$, HZ3$^{dd}$ & HZ3$^{c}$) with high resistivity signature of more than 1600 Ωm was
demarcated in all the 2D geoelectrical models of profile AA' projected by Wenner-
Schlumberger, dipole-dipole and combined inversion of both arrays at the surface distance of
about 195 to 270 m.
The soil/ alluvium layer showed low resistivity up to 10 m depth by both the techniques. A
signature of weather rock / fracture rock was delineated in least square inversion technique
only for dipole-dipole and combined inversion of both arrays. However, in robust inversion
technique this feature was visibly identified in all resistivity sections. The extension of water
aquifer zone at greater depth associated with fracture rock mass was well demarcated by
combined inversion of both arrays through L$_1$- norm in comparison to L$_2$-norm.
The outcomes generated of both synthetic and field conditions by inversion algorithm
revealed that a combination of Wenner-Schlumberger and dipole- dipole array would provide
maximum subsurface information and the optimal arrays sensitivity as this combination can
encompass both strong signal/noise ratio and sensitivity to vertical and lateral changes. A
prominent subsurface existing structure in geoelectrical sections by resistivity data sets could
be assessed by comparing the outcomes of inversion techniques. This is vital particularly
where sudden resistivity changes like geologic interfaces characterized by variation in
lithology are anticipated.
**4.0 Conclusions:**
Initially, the synthetic data was generated using Res2Dmod software. Field situation was
simulated through forward modelling. Two different algorithm techniques i.e. Least square
inversion and Robust inversion were studied in 2D electrical resistivity data set for mapping
of complex subsurface existing structures over a part of the CSIR-CIMFR campus using
Wenner-Schlumberger, dipole- dipole and combined inversion of both arrays. Robust
inversion indicates an additional feature with combined inversion of both arrays compared to
$L_2$-norm and it has good convergence throughout the iteration process, enabling easy
analysis. The extension of aquifer zone associated with fracture rock mass at greater depth
with high resolution was well demarcated by robust inversion indicates an additional feature
with combined inversion of both arrays. A complex subsurface existing structure in
geoelectrical sections by ERT data sets could be evaluated by comparing the consequences
from the two inversion schemes.
**5.0 Data availability:** Outcomes are in the form of images shown in Figs.1, 2, 3, 4, 5 and 6.
There is no data in addition.
**6.0 Team list:** Abhay Kumar Bharti, Amar Prakash and Krishna Kant Kumar Singh
**7.0 Author contribution:**
Abhay Kumar Bharti: Conducted field investigation, data interpretation and preparation of
manuscript.
Amar Prakash: Contributed in enhancing data interpretation and elevation in manuscript
quality.
Krishna Kant Kumar Singh: Contributed in site selection for investigation and data
interpretation.



**8.0 Competing interests:** Authors have not competing interest in any aspect.
**9.0 Disclaimer:** No such issue
**10. Acknowledgements:**
The authors extend thanks to Director, CSIR-Central Institute of Mining and Fuel Research,
Dhanbad, for providing relevant support, guidance in this study and permission for
publication.

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

281                                      **List of Figures**



**Figure.3:** Synthetic model outcomes (a) synthetic geological formation (b) inverted
resistivity model of Wenner–Schlumberger array (c) inverted resistivity model of dipole–
dipole array and (d) combined inversion of both arrays with Robust inversion technique
**Figure.4:** SYSCAL Pro-96 (Iris Instrument) data-acquisition field setup using 96 electrodes
**Figure.5:** 2D ERT section along profile AA' over the study area: (a) Wenner–Schlumberger
array, (b) dipole–dipole array and (c) combined inversion of both arrays with Least square
inversion technique
**Figure.6:** 2D ERT section along profile AA' over the study area: (a) Wenner–Schlumberger
array, (b) dipole–dipole array and (c) combined inversion of both arrays with Robust
inversion technique




















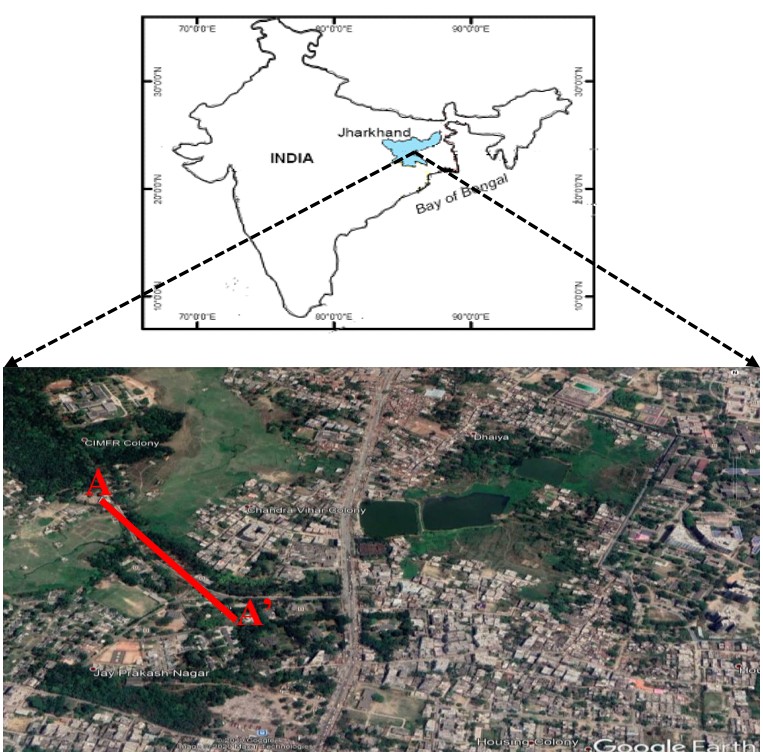

**Figure.1:** Location map of the study area © Google Earth





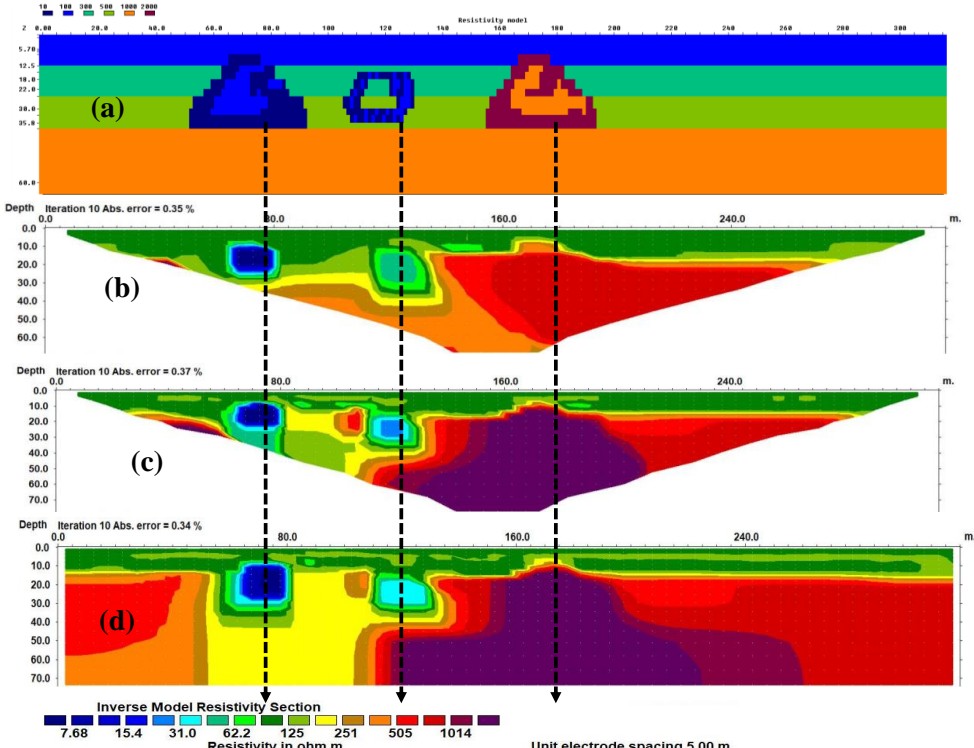

**Figure.2:** Synthetic model outcomes (a) synthetic geological formation (b) inverted resistivity model of Wenner–Schlumberger array (c) inverted resistivity model of dipole–dipole array and (d) combined inversion of both arrays with Least square inversion technique



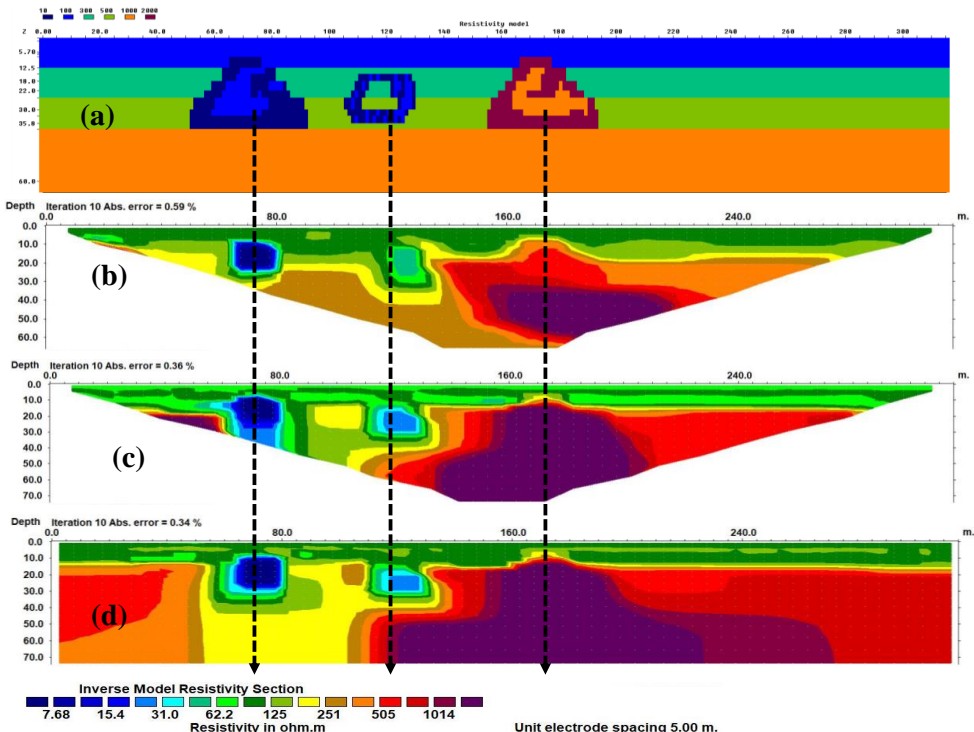

**Figure.3:** Synthetic model outcomes (a) synthetic geological formation (b) inverted resistivity model of Wenner–Schlumberger array (c) inverted resistivity model of dipole–dipole array and (d) combined inversion of both arrays with Robust inversion technique





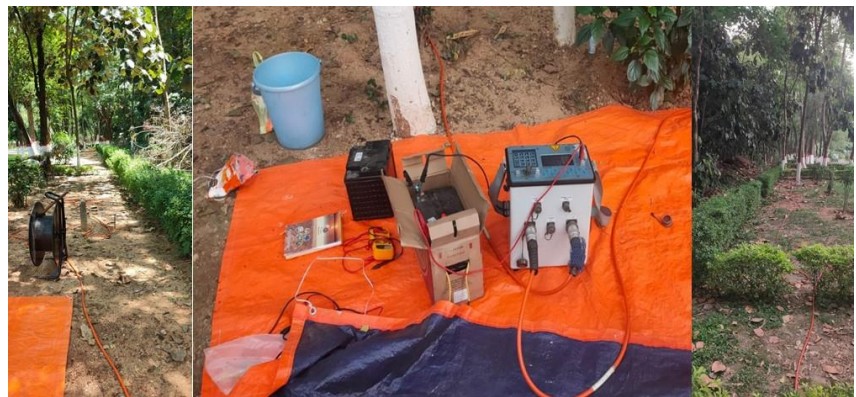

**Figure.4:** SYSCAL Pro-96 (Iris Instrument) data-acquisition field setup using 96 electrodes





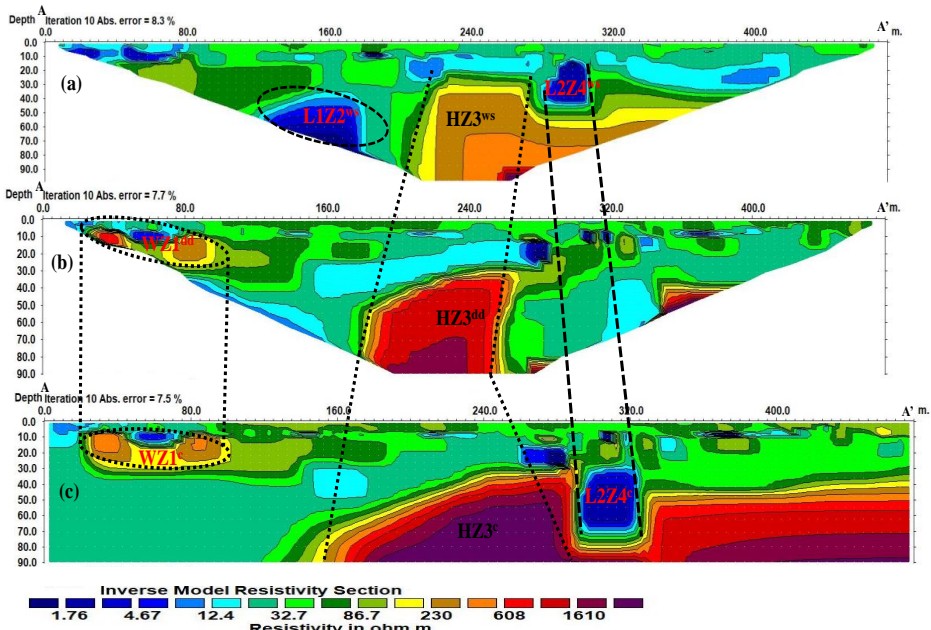

**Figure.5:** 2D ERT section along profile AA' over the study area: (a) Wenner–Schlumberger array, (b) dipole–dipole array and (c) combined inversion of both arrays with Least square inversion technique





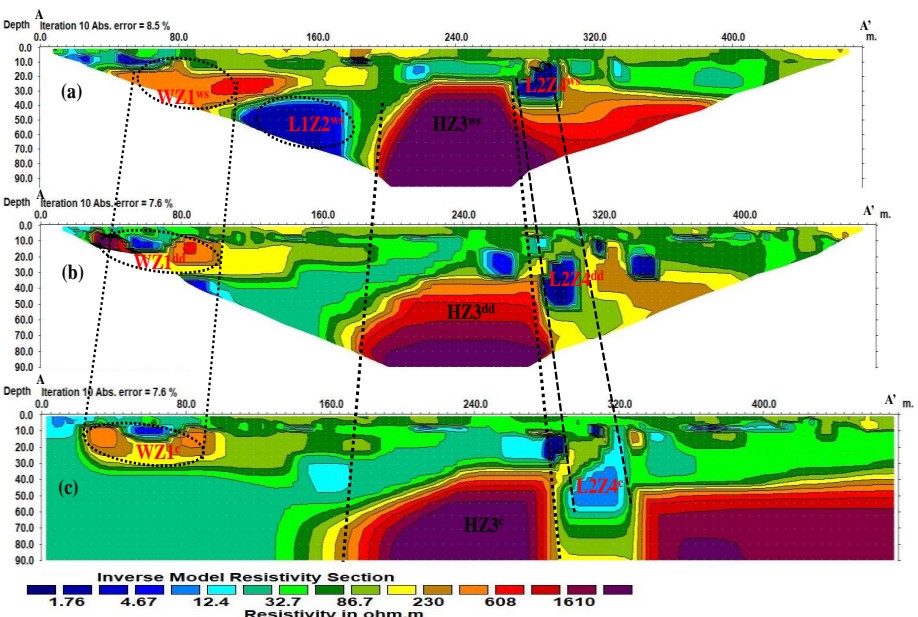

**Figure.6:** 2D ERT section along profile AA' over the study area: (a) Wenner–Schlumberger array, (b) dipole–dipole array and (c) combined inversion of both arrays with Robust inversion technique