# Peer review of "1.0 Introduction"

_Geoscientific Instrumentation, Methods and Data Systems, 2020_

## Referee Comment (RC1) · Anonymous Referee #1 · 13 Dec 2020

Authors try to compare the efficacy of two inversion approaches already available within the software package of RES2DINV (Loke and Barker 1996; Loke 1999). However, similar studies are already available in literature by the developer of the software (e.g., Loke et al., 2001; 2003). Thus, it was very hard to understand the importance and/or novelty of this work. Neither these aspects (including importance and objectives) have been mentioned in the introduction nor discussed anywhere in the manuscript. Also, the overall write up of the manuscript is VERY poor. Many places, I found continuities are missing, sentences are not proper, and meaning is incomplete. As a result, it is hard to follow the manuscript. Author should focus toward presenting a work dealing with enhancement of the technique or as comparison with

some global optimization techniques clearly highlighting the requirement/importance of the work or application of the existing methods on data from a new area as a new case study. I have mentioned some of these issues along with few technical points in the annotated pdf file as attached herewith. Hope these will be useful in improving the quality of the manuscript for future. Based on my observations, I suggest for Rejection of the manuscript.

Please also note the supplement to this comment:
https://gi.copernicus.org/preprints/gi-2020-25/gi-2020-25-RC1-supplement.pdf

―――――――――――――――――――――

---

## Referee Comment (RC2) · Anonymous Referee #2 · 31 Jul 2021

This paper address ERT numerical simulation, inversion schemes and application in field.

Anyway it is very difficult to identify the main innovative contribution of authors to this topic.

The numerical simulation part leans on available software for direct and and inversion schemes. Unfortunately there is conclusions does not show progress from existing works already published.

For the field application, the presented results are in phase with previous numerical

part, but no deeper analysis is proposed. Note that there is a problem with figures numbering in the text for this part that match numerical part and not field experiments part.

Globally this paper presents an application of existing methods available in an existing software but should have take benefit of deeper discussion on inversion schemes and possible improvement proposal . For the field experiments, results analysis should have been enhance by addressing a deeper quantitative analysis completed by tomographic view.

For all these reasons I suggest rejection of the manuscript

---

## Author Comment (AC1) · 31 Aug 2021

Anonymous Referee #1 Comment: Authors try to compare the efficacy of two inversion approaches already available within the software package of RES2DINV (Loke and Barker 1996; Loke 1999). However, similar studies are already available in literature by the developer of the software (e.g., Loke et al., 2001; 2003). Thus, it was very hard to understand the importance and/or novelty of this work. Neither these aspects (including importance and objectives) have been mentioned in the introduction nor discussed anywhere in the manuscript. Also, the overall write up of the manuscript is VERY poor. Many places, I found continuities are missing, sentences are not proper,

and meaning is incomplete. As a result, it is hard to follow the manuscript. Author should focus toward presenting a work dealing with enhancement of the technique or as comparison with some global optimization techniques clearly highlighting the requirement/importance of the work or application of the existing methods on data from a new area as a new case study. I have mentioned some of these issues along with few technical points in the annotated pdf file as attached herewith. Hope these will be useful in improving the quality of the manuscript for future. Based on my observations, I suggest for Rejection of the manuscript.

Reply: Comments are welcome. The manuscript has been modified extensively. There are hardly a few works on compare the efficacy of two inversion approaches for poised mapping of subsurface features using electrical resistivity tomography survey. Initially, the inversion algorithm techniques are employed on synthetic model data set with and without some random Gaussian noise, and its validity is tested by field data set using Wenner-Schlumberger, Dipole-Dipole and joint inversion of both arrays. Smoothness Constrained Least-squares technique (L2-norm) was used for inversion by handling software RES2DINV (Sasaki, 1992; Loke, 1997; Loke and Barker, 1996). In this inversion technique, the subsurface is divided into number of rectangular blocks of constant resistivity. Then the resistivity of each block is evaluated by minimizing the difference between observed and calculated pseudo-sections using an iterative scheme. The smoothness-constraint leads the algorithm to yield a solution with smooth resistivity variations.

Further, L1-norm regularization inversion technique (called also robust or blocky method) was used for inversion. This technique minimizes the absolute differences between measured and calculated apparent resistivity values by an iterative process (Loke et al., 2003; Wolke & Schwetlick, 1988), in which the accuracy of the data fit is expressed in terms of the absolute error (Claerbout and Muir, 1973). The calculated pseudo-sections could be achieved by either finite-difference or finite-element methods (Coggon 1971; Dey and Morrison 1979).

The outcomes generated of both synthetic and field conditions by inversion algorithm revealed that a combination of Wenner-Schlumberger and dipole- dipole array would provide maximum subsurface information and the optimal arrays sensitivity as this combination can encompass both strong signal/noise ratio and sensitivity to vertical and lateral changes. A prominent subsurface existing structure in geoelectrical sections by resistivity data sets could be assessed by comparing the outcomes of inversion techniques. This is vital particularly where sudden resistivity changes like geologic interfaces characterized by variation in lithology are anticipated.

References: Claerbout J F and Muir F 1973 Robust modeling with erratic data. Geophysics, 38: 826–844. Loke M H, Acworth I and Dahlin T 2003 A comparison of smooth and blocky inversion methods in 2D electrical imaging surveys; Explor. Geophys. 34, 182–187. Wolke R and Schwetlick H 1988 Iteratively reweighted least-squares algorithms, convergence analysis, and numerical comparisons; SIAM Journal of Scientific and Statistical Computations, 9: 907–921.

---

## Author Comment (AC2) · 31 Aug 2021

Anonymous Referee #2 Comment: This paper address ERT numerical simulation, inversion schemes and application in field. Anyway, it is very difficult to identify the main innovative contribution of authors to this topic. The numerical simulation part leans on available software for direct and inversion schemes. Unfortunately, there is conclusions does not show progress from existing works already published. For the field application, the presented results are in phase with previous numerical part, but no deeper analysis is proposed. Note that there is a problem with figures numbering in the text for this part that match numerical part and not field experiment's part. Globally this paper

presents an application of existing methods available in an existing software but should have taken benefit of deeper discussion on inversion schemes and possible improvement proposal. For the field experiments, results analysis should have been enhanced by addressing a deeper quantitative analysis completed by tomographic view. For all these reasons I suggest rejection of the manuscript.

Reply: Comments are welcome. The manuscript has been modified extensively. There are hardly a few works on compare the efficacy of two inversion approaches for poised mapping of subsurface features using electrical resistivity tomography survey. Initially, the inversion algorithm techniques are employed on synthetic model data set with and without some random Gaussian noise, and its validity is tested by field data set using Wenner-Schlumberger, Dipole-Dipole and joint inversion of both arrays. Smoothness Constrained Least-squares technique (L2-norm) was used for inversion by handling software RES2DINV (Sasaki, 1992; Loke, 1997; Loke and Barker, 1996). In this inversion technique, the subsurface is divided into number of rectangular blocks of constant resistivity. Then the resistivity of each block is evaluated by minimizing the difference between observed and calculated pseudo-sections using an iterative scheme. The smoothness-constraint leads the algorithm to yield a solution with smooth resistivity variations.

Further, L1-norm regularization inversion technique (called also robust or blocky method) was used for inversion. This technique minimizes the absolute differences between measured and calculated apparent resistivity values by an iterative process (Loke et al., 2003; Wolke & Schwetlick, 1988), in which the accuracy of the data fit is expressed in terms of the absolute error (Claerbout and Muir, 1973). The calculated pseudo-sections could be achieved by either finite-difference or finite-element methods (Coggon 1971; Dey and Morrison 1979).

The outcomes generated of both synthetic and field conditions by inversion algorithm revealed that a combination of Wenner-Schlumberger and dipole- dipole array would provide maximum subsurface information and the optimal arrays sensitivity as this combination can encompass both strong signal/noise ratio and sensitivity to vertical and lateral changes. A prominent subsurface existing structure in geoelectrical sections by resistivity data sets could be assessed by comparing the outcomes of inversion techniques. This is vital particularly where sudden resistivity changes like geologic interfaces characterized by variation in lithology are anticipated.

References: Claerbout J F and Muir F 1973 Robust modeling with erratic data. Geophysics, 38: 826–844. Loke M H, Acworth I and Dahlin T 2003 A comparison of smooth and blocky inversion methods in 2D electrical imaging surveys; Explor. Geophys. 34, 182–187. Wolke R and Schwetlick H 1988 Iteratively reweighted least-squares algorithms, convergence analysis, and numerical comparisons; SIAM Journal of Scientific and Statistical Computations, 9: 907–921.